# Ingenious Fabrication of Ag-Filled Porous Anodic Alumina Films as Powerful SERS Substrates for Efficient Detection of Biological and Organic Molecules

**DOI:** 10.3390/bios12100807

**Published:** 2022-09-29

**Authors:** Chih-Yi Liu, Rahul Ram, Rahim Bakash Kolaru, Anindya Sundar Jana, Annada Sankar Sadhu, Cheng-Shane Chu, Yi-Nan Lin, Bhola Nath Pal, Shih-Hsin Chang, Sajal Biring

**Affiliations:** 1Organic Electronics Research Center, Ming Chi University of Technology, New Taipei City 24301, Taiwan; 2Department of Electronic Engineering, Ming Chi University of Technology, New Taipei City 24301, Taiwan; 3Department of Mechanical Engineering, Ming Chi University of Technology, New Taipei City 24301, Taiwan; 4School of Material Science and Technology, Indian Institute of Technology, BHU, Varanasi 221005, India; 5MSSCORPS Co., Ltd., Hsinchu 300047, Taiwan

**Keywords:** surface-enhanced Raman scattering, SERS, silver nanoparticles, anodic alumina, organic molecule detection, biomolecule detection

## Abstract

Surface-enhanced Raman scattering (SERS) has been widely used to effectively detect various biological and organic molecules. This detection method needs analytes adsorbed onto a specific metal nanostructure, e.g., Ag-nanoparticles. A substrate containing such a structure (called SERS substrate) is user-friendly for people implementing the adsorption and subsequent SERS detection. Here, we report on powerful SERS substrates based on efficient fabrication of Ag-filled anodic aluminum oxide (AAO) films. The films contain many nanopores with small as-grown inter-pore gap of 15 nm. The substrates are created by electrochemically depositing silver into nanopores without an additional pore widening process, which is usually needed for conventional two-step AAO fabrication. The created substrates contain well-separated Ag-nanoparticles with quite a small inter-particle gap and a high number density (2.5 × 10^10^ cm^−2^). We use one-step anodization together with omitting additional pore widening to improve the throughput of substrate fabrication. Such substrates provide a low concentration detection limit of 10^−11^ M and high SERS enhancement factor of 1 × 10^6^ for rhodamine 6G (R6G). The effective detection of biological and organic molecules by the substrate is demonstrated with analytes of adenine, glucose, R6G, eosin Y, and methylene blue. These results allow us to take one step further toward the successful commercialization of AAO-based SERS substrates.

## 1. Introduction

With its ability to identify chemical bonding, Raman spectroscopy has been widely used to sense structural fingerprints of various materials [1,2]. However, this technique suffers from low sensitivity and thus shows a strong limitation in detecting small amounts of analytes, e.g., single-molecule investigation. This drawback has been strongly improved by using surface-enhanced Raman scattering (SERS), which is a phenomenon of strongly increased Raman signals when corresponding analytes are adsorbed or close to a specific metal nanostructure, such as silver nanoparticles [3,4,5,6,7,8,9,10]. Nowadays, SERS technology has been widely employed to sense various biomolecules [11,12,13]. For example, Wang’ group has developed the technology of rapid bacterial antibiotic susceptibility tests by using SERS to investigate adenine concentration, which is an indicator of bacterium survival [14].

The key factor of SERS technology is to fabricate a proper metal nanostructure with high Raman enhancing power. It is easy to create such a structure by using a porous anodic alumina oxide (AAO) film [15,16,17,18,19,20]. This film contains packed nanopores that can be filled with different materials to produce various nanostructures [21,22,23,24]. Growing silver into the nanopores of an AAO film creates an SERS substrate (called Ag-AAO SERS substrate hereafter) containing a high density of Ag-nanoparticles [25,26,27,28,29,30]. Such a substrate usually requires a nanoparticle size of a few tens of nm and an inter-particle gap as small as possible to achieve a higher SERS enhancement factor (EF). In addition, this factor also increases with an increase in Ag-nanoparticle density. AAO films used for SERS substrates are often grown using a time-consuming two-step anodization in an electrolyte of sulfuric or oxalic acid [31,32,33]. The first step usually takes several hours to create the long straight nanopores, whose bottom ends (AAO/Al interface) self-organize into a certain degree of ordered structure. Removing the AAO grown in the first step leaves ordered concaves in the residual Al surface. The concaves guide the growth of nanopores to form an ordered structure in the following second step of anodization. In addition, the gaps between as-grown AAO nanopores being ~50% larger than their diameter is unfavorable to SERS application. Therefore, an additional pore widening process implemented by chemical etching is usually needed to reduce the gap between the nanopores of AAO films for creating substrates with high SERS EF. Here, we report on new types of Ag-AAO SERS substrates that are fabricated using one-step anodization together with omitting the pore widening process. This fabrication is able to reduce AAO creation time from several hours to 30 min and is thus more efficient.

This fabrication uses an electrolyte of phosphoric acid and an anodization voltage of 10 V. Phosphoric acid is usually employed for growing AAO with a pore size and gap both larger than 200 nm and is thus barely applied to fabricate SERS substrates within the current literature. However, the voltage used here can overcome this drawback. The fabricated AAO is composed of serrated nanopores [34,35] with a small inter-pore gap of 15 nm in as-grown AAO, and, thus, an additional pore widening process can be omitted. In addition, the anodization voltage for creating SERS substrates in sulfuric (~20 V) or oxalic (~40 V) acid is usually larger than that of our fabrication. This difference may cause a higher nanopore density created in the fabrication process presented here, because the inter-pore distance increases with anodization voltage, irrespective of electrolytes [36,37,38]. Growing silver into highly dense nanopores creates more Ag-nanoparticles and thus more SERS enhancers. More enhancers cause an increase in signal, which, in turn, leads to the facilitation of SERS detection. The SERS substrate presented here was tested with analytes of adenine, glucose, rhodamine 6G (R6G), eosin Y, and methylene blue. All the testing cases clearly provided enhanced Raman signals. In other words, this new fabrication method is able to create powerful SERS substrates with a higher throughput.

## 2. Materials and Methods

### 2.1. Electropolishing of Al Foils

Aluminum foils (size of 2 cm × 2 cm, thickness of 0.2 mm, and purity of 99.99%) were electropolished in a mixture of ethanol (1300 mL) and HClO_4_ solution. The HClO_4_ solution was prepared by mixing 188 mL of HClO_4_ with 72 g of DI water. A stainless-steel plate was used as cathode, while the aluminum foil to be polished was used as anode. The electrolyte was maintained at 5 °C with a fine stirring rod. A current density of 0.16 A/cm^2^ was applied initially for fast polishing of the rough sample surface with a circular area of 1 cm^2^. After reaching 20 V, the input voltage was reduced to 5 V for fine polishing and was kept constant for 25 min.

### 2.2. Fabrication of Ag-AAO SERS Substrates

Polished Al foils were used to fabricate Ag-AAO SERS substrates as schematically shown in Appendix A. The foils were first anodized in phosphoric acid of 6% at 10 V with a stainless-steel plate as a cathode for 30 min at a temperature of 22 °C. Silver electrodeposition was then carried out by using an electrolyte containing 41 g of MgSO_4_ and 1 g of AgNO_3_ dissolved in 1 L of water. In addition, 1.8 mL of H_2_SO_4_ (98%) was also added into the solution. The solution was maintained at 31 °C during the deposition process. A 60 Hz AC voltage of 9 V with different deposition cycles was applied to electrochemically grow Ag nanoparticles inside the nanopores; each cycle contained 0.25 s power-on and 4.75 s power-off time. After deposition, the samples were dipped into 0.2 M HCl for 20 s and then cleaned with DI water.

### 2.3. Raman Measurement

Prior to Raman measurement, fabricated Ag-AAO SERS substrates were dipped into analyte solutions for 2 min and then dried by spin casting with 2000 rpm for 20 s. The analyte species used in this study were adenine, glucose, R6G, eosin Y, and methylene blue. The analyte molecules were adsorbed onto the substrates after the drying process. The Raman signals from the adsorbed molecules were used to identify the SERS enhancement power of the corresponding substrate. Raman measurement was performed on a micro-Raman system with excitation light sources of a He-Ne (LASOS Laser GmbH, wavelength of 633 nm) and Nd:YAG (AST Instruments corp. wavelength of 532 nm) laser. After passing through a narrow bandpass filter to remove residual plasma lines, a selected laser beam (wavelength of 532 or 633 nm) was focused by a 10× (NA = 0.25) or 50× (NA = 0.50) objective lens (Olympus Corp., Tokyo, Japan) to a sample. The scattered radiation was collected by the same objective lens and then sent through a Raman edge filter to a monochromator (iHR 550, Horiba, Kyoto, Japan) with a grating of 1200 groves/mm. The system was controlled by a computer, and the detected spectra were displayed on a commercial monitor.

## 3. Results and Discussion

### 3.1. Fabrication of Anodic Alumina 

We selected a fixed low anodization voltage of 10 V to grow AAO films because their short inter-nanopore distance can be used to create Ag-nanoparticles with a small gap without an additional pore widening process. Appendix A shows the plot of reaction current as a function of time in a typical anodization process. The current reaches 23 mA sharply in the initial anodization period and then gradually falls to 10 mA. After that, the current increase again to 13 mA within 2 min and stabilizes in the subsequent process. The AAO nanopores were grown continuously at this stable current.

Figure 1a,b show the topside SEM micrographs of AAO films grown with an anodization time of 5 and 30 min, respectively. Both figures display nanopores with a number density of 2.5 × 10^10^ cm^−2^. Such dense nanopores can be used to create a high density of Ag nanoparticles, facilitating the fabrication of powerful SERS substrates. The size of the nanopores restricts the diameter of the subsequently fabricated Ag-nanoparticles. The pore size in Figure 1a is dramatically smaller than that in Figure 1b. The difference originates from the chemical etching of aluminum oxide in H_3_PO_4_ electrolytic solution. A longer anodization time leads to more oxide etching and thus causes a more porous structure as observed by comparing Figure 1a,b. Such etching enlarges the diameter and reduces the oxide wall of nanopores in the anodization process. Nanoparticles grown in such nanopores with a thin oxide wall offer a small inter-particle gap. Such gaps are advantageous for creating powerful SERS substrates, which usually require metal nanoparticles with small intervals. In other words, the AAO films with a thickness of 1 μm (Figure 1c) presented here could be used to fabricate effective SERS substrates without an additional pore widening process.

### 3.2. Characterization of Ag-AAO SERS Substrate

SERS substrates were fabricated by AC electrodeposition of silver into AAO nanopores with an anodization time of 30 min. A commercial dual-beam focused ion beam was used to expose the cross-sections of the substrates for interior investigation. Figure 2a–c show the SEM micrographs of the cross-sections with different deposition cycles of 6, 13, and 39. Silver was deposited into nanopores to form nanoparticles as shown by the white spots in the micrographs. The lateral size of Ag-nanoparticles for the 13-cycle case (Figure 2b) is similar to the 39-cycle case (Figure 2c) because of the confinement caused by nanopores with a diameter of ~25 nm. Deposited material for six cycles seems insufficient, and, thus, the lateral diameter of Ag-nanoparticles for Figure 2a is somewhat smaller.

Without growth confinement in a vertical direction, the height of Ag-nanoparticles clearly increases with the cycle number as shown in Figure 2; the heights caused by 6, 13, and 39 cycles are 20 (Figure 2a), 30 (Figure 2b), and 70 (Figure 2c) nm, respectively. However, the vertical growth does not rise linearly with the cycle number; its average rate for 6 cycles is ~2 times of that for 39 cycles. Such non-linear growth may originate from the concentration of electrolytes decreasing with the increasing electro-chemical deposition, because the AC reaction probably provides insufficient time to resupply consumed reacting species from the bulk solution. Nevertheless, the confinement of nanopores causes asymmetrical growth of silver, and, thus, nanoparticles convert to nanowires after long-term electro-chemical deposition.

The composition of SERS substrates was investigated with EDS. Table 1 shows the composition ratio of aluminum, oxygen, and silver for a substrate with 13 cycles of deposition. The atomic concentration of Al (48.79%) is close to that of O (49.68%), indicating that the ratio of the two elements is ~1:1. In general, a stable aluminum oxide (Al_2_O_3_) contains a 2:3 atomic ratio of Al to O. The higher Al ratio may originate from the additional EDS signal contributed by the unanodized aluminum under AAO. Nevertheless, the EDS investigation displayed in Table 1 demonstrates that SERS substrates are mainly composed of Al, O, and Ag.

Electromagnetic (EM) enhancement in SERS is contributed by the plasma oscillation of free electrons in metal, which is maximized, while the frequency of the incident matches that of plasma resonance oscillation. The plasmon resonance frequency or the wavelength of the SERS substrate could be measured by capturing the scattering spectrum under white light illumination. Figure 3 displays a scattering spectrum from a typical SERS substrate with 13 deposition cycles. The spectrum shows a peak at 518 nm, which reflects the plasmon resonance wavelength of Ag-nanoparticles embedded in the AAO substrate. Therefore, an excitation light with a wavelength close to the plasmon resonance wavelength should lead to strong SERS signals, which is discussed in detail below.

### 3.3. SERS Measurement

We used 10^−4^ M adenine as a probe to investigate the efficiency of SERS substrates, as shown in the spectra in Figure 4. Two excitation lasers with different wavelengths of 532 nm and 633 nm were used for the investigation. The spectra show a peak at 734 cm^−1^ (purine ring breathing mode) in both cases. However, the Raman intensity for the 532 nm case is 7 times higher than that at the 633 nm case. The reason causing this huge difference lies in the scattering spectra in Figure 3, which shows a resonance peak at 518 nm. The excitation wavelength of 532 nm compared to 633 nm is much closer to the resonance peak and thus provides a higher Raman enhancement power in SERS measurements. Consequently, 532 nm is always used as the excitation wavelength in the following SERS studies.

It is reasonable to assume that the SERS enhancement power of substrates is highly dependent on their silver morphology, which is dominated by the Ag-growth condition. The condition for the maximum enhancement can be found via a systematic study of Ag-growth cycles, as shown in Figure 5. We used R6G solution of 10^−5^ M as a probe for this study. Several Raman peaks of R6G can be observed in the SERS spectra presented in Figure 5a. The most prominent peak at 613 cm^−1^ (in-plane xanthene ring deformations) was used to estimate the SERS enhancement power of the corresponding substrates. Peak intensities of 613 cm^−1^ vary with different growth cycles. We used this peak as an indicator to quantitatively plot signal intensities as functions of Ag-deposition cycles, as shown in Figure 5b. This figure clearly indicates that the substrate with Ag deposited at 13 cycles provides the highest SERS peak intensity and thus the maximum enhancement power. In other words, SERS intensity is sensitive to deposition cycles. For example, the intensity of a 13-cycle case (26,000 counts) is 11 times higher than that of a 6-cycle case (2400 counts) and 13 times higher than that of a 39-cycle case (2000 counts). These huge differences indicate that Ag-growth conditions need to be carefully selected for fabricating substrates with high SERS EF.

In addition to EF, the lowest detection limit of analyte concentrations is another crucial factor to judge the quality of SERS substrates. Figure 6a shows the SERS spectra of R6G with varied concentrations. Several Raman peaks of R6G can be observed clearly in the spectra. The most prominent peak of 613 cm^−1^ was again used to identify the detection limit of the analyte molecules. This peak can be clearly observed at any concentration between 10^−11^ and 10^−5^ M, as shown in Figure 6a. We used this peak as an indicator to quantitatively plot signal intensities as functions of Ag-deposition cycles, as shown in Figure 6b. The intensity gradually increases from 900 to 78,000 counts, while the concentration rises from 10^−11^ to 10^−5^ M. The inset in Figure 6b represents an enlarged plot at lower concentrations to estimate the limit of detection. This study indicates that the Ag-AAO substrate could even detect R6G at an extremely low concentration of 10^−11^ M without any ambiguity. This result demonstrates the low detection limit of the proposed SERS substrate. These substrates also provide high spatial uniformity as displayed by the error bars shown in Figure 6b. Each error bar is calculated from the standard deviation of 613-cm^−^^1^ peaks in 48 spectra based on three arbitrarily selected Raman mappings of 4×4 square lattices with a lattice constant of 40 μm. For example, the relative standard deviation (standard deviation divided by signal intensity) for 10^−5^ M is 8%, which indicates a high spatial uniformity. Appendix A shows the three corresponding mappings for the concentration of 10^−5^ M, which directly display the uniformity of the substrate. The combined results shown in Figure 6a,b indicate that the proposed SERS substrates provide both a low detection limit and a high spatial uniformity.

Ag-AAO SERS substrates can be used to effectively detect many biological and organic molecules. To demonstrate the detection ability, we used three different analytes of eosin Y, adenine, and methylene blue in addition to R6G to probe the SERS efficiency of Ag-AAO substrates. Figure 7a shows the SERS spectra of eosin Y solutions with concentrations of 10^−6^, 10^−7^, and 10^−9^ M. The characteristic Raman peaks of eosin Y can be observed in the spectra even for the low concentration case of 10^−9^ M. Figure 7b displays SERS spectra of adenine solutions with concentrations of 10^−4^, 10^−5^, 10^−6^ M, and 10^−7^ M. All the spectra show a clear adenine peak at 734 cm^−1^, revealing that the detection limit of adenine is much lower than 10^−7^ M. The SERS spectra for methylene blue solutions with concentrations of 10^−5^, 10^−6^, and 10^−7^ M are shown in Figure 7c. All the spectra display characteristic Raman peaks of methylene blue. Appendix A shows the plot of peak intensities as functions of analyte concentrations based on the spectra in Figure 7. The data in Figure 7 and Appendix A unequivocally demonstrate the detection ability of the SERS substrate presented here. Appendix A shows the SERS spectra of glucose with concentrations of 10^−3^, 10^−4^, and 10^−5^ M. It is relatively difficult to detect glucose molecules with SERS technology because of their small Raman cross-section and low affinity to bind with a metal nanostructure [39]. The peak of 912 cm^−1^ (corresponding to C-OH stretching) being observed in all three spectra in Appendix A indicates the glucose detection limit reaching 10^−5^ M. The combined results of Figure 6, Figure 7, Appendix A leads to the conclusion that the Ag-AAO SERS substrate presented here is able to effectively detect biological and organic molecules. Appendix A shows the Raman spectra of 10^−3^ M of R6G (Appendix A), eosin Y (Appendix A), and adenine (Appendix A) without SERS substrates. No Raman peak can be observed for all three spectra, indicating that SERS substrates are indeed required for sensing analyte molecules.

The enhancement power of SERS substrates was quantified with their EFs based on the following equation [40]:EF = (N_Reference_ × I_SERS_)/(N_SERS_ × I_Reference_)(1)
where N_Reference_ and I_Reference_ are the number of molecules and the Raman signal of a reference analyte, respectively. N_SERS_ and I_SERS_ are the number of molecules and Raman signal from an analyte on Ag-AAO SERS substrate, respectively. We used R6G solution as an analyte for the EF estimation. I_Reference_ was measured from the focal spot (90 μm diameter) of the excitation laser focused inside a solution of 0.1 M R6G. The diameter of the spot was estimated by focusing the excitation laser light on a silicon substrate under the same conditions. As a result, we obtained an estimated N_Reference_ of 2.4 × 10^13^ and I_Reference_ of 2400 counts for R6G analyte. The SERS measurement of 10^−5^ M R6G (Figure 6b) was used to estimate N_SERS_ and I_SERS_. We assumed only the first mono-layer coated on Ag-nanoparticles providing an effective SERS signal. The effective SERS-detecting area of a single Ag-nanoparticle is 900 nm^2^ (assumed as half of a sphere with radius of 12 nm), while the size of the R6G molecule is estimated as 2 nm^2^. The Ag-nanoparticle number (density of 2.5 × 10^10^ cm^−2^) in the excitation light spot (diameter of 90 μm) is 1.6 × 10^6^. Therefore, N_SERS_ = (900/2) × 1.6 × 10^6^ ~ 7.2 × 10^8^. Figure 6b points out the I_SERS_ of 78,000 counts. The above estimated parameters are substituted into Equation (1) to obtain EF of ~1 × 10^6^.

## 4. Conclusions

AAO-based SERS substrates have many advantages, such as high uniformity, large area, facile fabrication and operation, and a powerful detection ability. The porous AAOs employed for the substrates are often fabricated via a time-consuming, two-step anodization together with an additional pore widening process. To improve fabrication efficiency, we developed a new type of Ag-AAO SERS substrates based on a one-step anodization method without additional pore widening. Anodization is performed in phosphoric acid, which is able to etch aluminum oxide and thus gradually enlarge the nanopores of AAO in their growth process. In addition, this etching also reduces the thickness of the oxide wall between the nanopores. The nanopores are then electrochemically filled by silver to create a high density of Ag-nanoparticles. The reduced wall thickness leads to a small inter-nanoparticle gap, which is favorable to SERS detection, and, thus, additional pore widening is unnecessary. The new fabrication recipe reduces the AAO creation time from several hours to 30 min. In other words, it is a more efficient process and thus provides a benefit in mass production with high throughput. The substrates presented here provide an SERS enhancing factor of 1 × 10^6^ and a detection limit of 10^−11^ M based on R6G sensing. We also demonstrated their ability to detect biological and organic molecules by testing with analyte solutions of adenine, glucose, R6G, eosin Y, and methylene blue. Appendix A summarizes the recent development of AAO-based SERS substrates [41,42,43,44,45,46,47,48,49,50,51]. According to the detection limits presented in this table, the AAO-based SERS substrate reported here shows a promising Raman enhancement performance. With their efficient production, high enhancing factor, low detection limit, and multi-species detection ability, the new-type Ag-AAO SERS substrates have great potential in biological and organic sensing in the future. 

## Figures and Tables

**Figure 1 biosensors-12-00807-f001:**
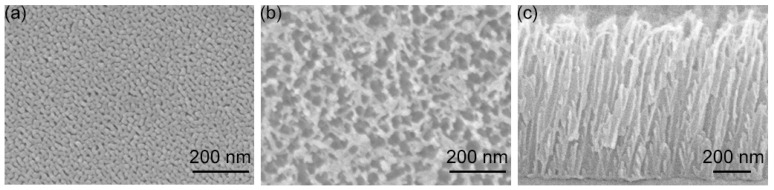
Top view SEM micrographs of AAO films after anodization time of (**a**) 5 and (**b**) 30 min. (**c**) Cross-section SEM image of the sample for (**b**) with a view angle of 38 degrees from the topside.

**Figure 2 biosensors-12-00807-f002:**
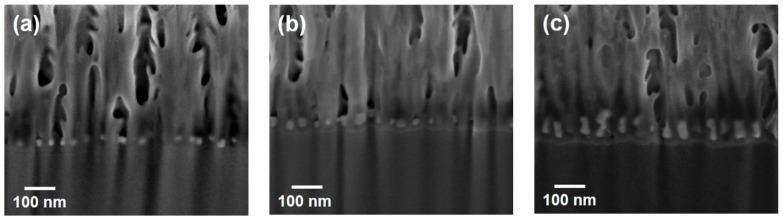
Typical cross-sectional SEM micrographs of Ag-AAO substrates with Ag-deposition cycles of (**a**) 6, (**b**) 13, and (**c**) 39.

**Figure 3 biosensors-12-00807-f003:**
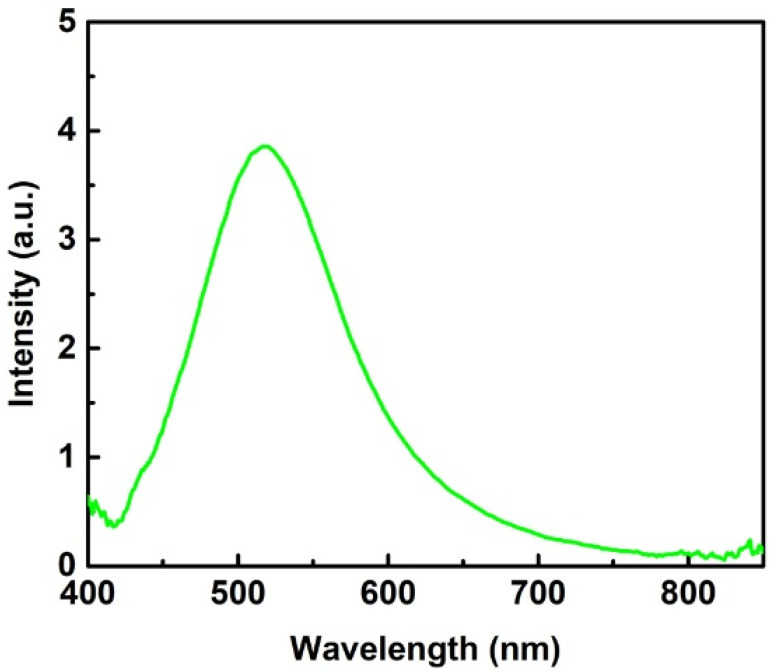
Scattering spectrum of a typical SERS substrate with 13 deposition cycles.

**Figure 4 biosensors-12-00807-f004:**
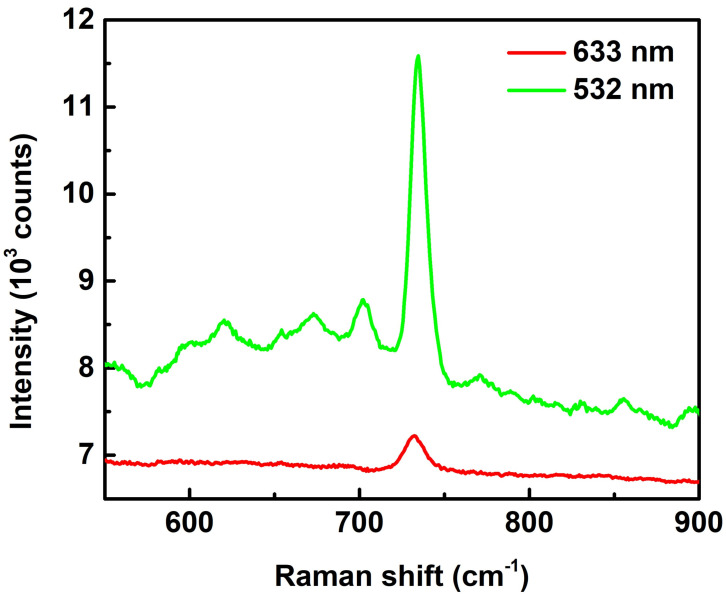
SERS spectra of adenine (concentration of 10^−4^ M) with excitation laser wavelengths of 633 and 532 nm. The spectra are pristine with neither intensity shift nor background subtraction. The objective lens of 50× and exposure times of 3 s were used for the measurements.

**Figure 5 biosensors-12-00807-f005:**
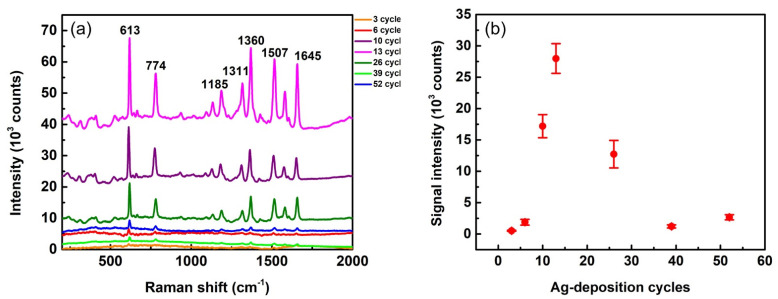
(**a**) SERS spectra from R6G (concentration of 10^−5^ M) on Ag-AAO substrates with different cycles of Ag deposition. The spectra are pristine with neither intensity shift nor background subtraction. (**b**) Signal intensity of the 613 cm^−1^ peak vs. number of Ag-deposition cycles. The laser wavelength of 532 nm, lens of 10×, and exposure times of 1 s were used for the measurement.

**Figure 6 biosensors-12-00807-f006:**
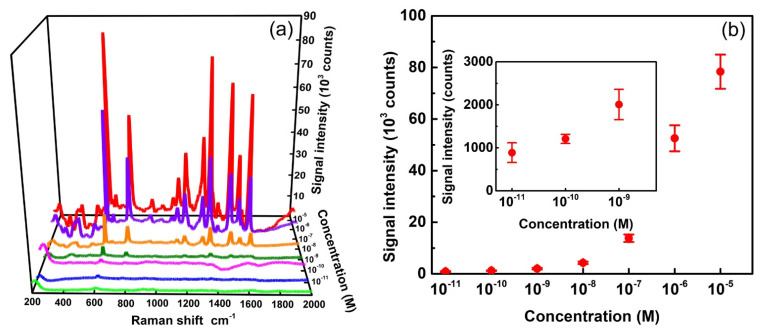
(**a**) SERS spectra of R6G with concentrations from 10^−5^ M to 10^−11^ M on Ag-AAO substrates. The spectra are pristine with neither intensity shift nor background subtraction. (**b**) SERS signal intensity at 613 cm^−1^ as a function of the R6G concentration. The laser wavelength of 532 nm, objective lens of 10×, and exposure times of 3 s were used for the measurement. Inset shows a zoomed-in area of the plot from concentrations between 10^−11^ M and 10^−9^ M.

**Figure 7 biosensors-12-00807-f007:**
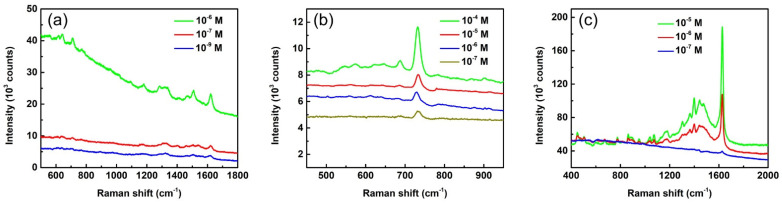
SERS spectra of (**a**) eosin Y, (**b**) adenine, and (**c**) methylene blue with different concentrations. The spectra are pristine with neither intensity shift nor background subtraction. The laser wavelength of 532 nm and lens of 50× were used for the measurement. The exposure times are 1 s for (**a**), 3 s for (**b**), and 10 s for (**c**).

**Table 1 biosensors-12-00807-t001:** Composition of SERS substrate with 13 deposition cycles determined via EDS investigation.

Element	Atomic Concentration (%)	Weight Concentration (%)
Aluminum	48.79	57.84
Oxygen	49.68	34.93
Silver	1.52	7.23

## Data Availability

Not applicable.

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
