# Peer review of "Ingenious Fabrication of Ag-Filled Porous Anodic Alumina Films as Powerful SERS Substrates for Efficient Detection of Biological and Organic Molecules"

_biosensors, 2022, doi:10.3390/bios12100807_

Round 1

Reviewer 1 Report

This paper described a powerful AAO-based SERS substrate for biological detection. The data and results on the preparation and characterization of the AAO were abundant. However, some concerns still should be addressed carefully before published. Following are my comments:

1. The authors tested lots of biological and organic molecules with different concentrations. However, for biological detection, the control group also needs to be tested to verify its performance and detection limit. There was no control group in each test (e.g. R6G, Eosin Y, and adenine).

2. The performance comparison between this proposed SERS substrate and some other common SERS materials should be summarized to prove that this AAO-based SERS substrate has great Raman enhancement performance.

3. To confirm the ability of biological detection of Eosin Y, adenine and methylene blue using this proposed Ag-AAO SERS substrate, the relationship between the signal intensity and concentration is suggested to be described and draw figures like Figure 5b.

4.  In Figure 6a, the vertical axis title is missing.

5. Most of the literature cited in this article are before 2018 (28 in total), authors should review and cite more new papers published in recent years.

6. The units in the full test need to be unified. (e.g. min/minutes)

Reviewer 2 Report

Sajal Biring et. al. introduce a novel method to prepare Ag-filled AAO films. I will recommend this paper after the following questions are resolved:

1. High uniformity is a major advantage for AAO-based SERS substrates. Authors should provide the SERS imaging results to demonstrate the uniformity of their substrates.

2. In Figure 6b, the SERS signals as an exponent-increasing-like function of the R6G concentration. However, the relation between SERS intensity and analyte concentration has to be followed with adsorption models like Langmuir isotherms. 

3.  Based on Figure 6b, it is not a mono-layer condition at 10-5 M R6G. Authors can not use the SERS signal at  10-5 M R6G to estimate the enhancement factor.

Round 2

Reviewer 1 Report

The author has given good answers and revisions to reviewer's comments. And the revisied article meets reviewer's requirements. Thus, I suggest it to be published in Biosensors.